# Effect of Nitrogen Application Rate on the Relationships between Multidimensional Plant Diversity and Ecosystem Production in a Temperate Steppe

**DOI:** 10.3390/biology13080554

**Published:** 2024-07-23

**Authors:** Gossaye Hailu Debaba, Kunyu Li, Xiaowei Wang, Yanan Wang, Wenming Bai, Guoyong Li

**Affiliations:** 1International Joint Research Laboratory for Global Change Ecology, School of Life Sciences, Henan University, Kaifeng 475004, China; gossaye@henu.edu.cn (G.H.D.); 104753170857@vip.henu.edu.cn (K.L.); wangxw0331@henu.edu.cn (X.W.); wangyanan@henu.edu.cn (Y.W.); 2State Key Laboratory of Vegetation and Environmental Change, Institute of Botany, Chinese Academy of Sciences, Beijing 100093, China; bwming@ibcas.ac.cn

**Keywords:** biodiversity and ecosystem functioning (BEF), species richness, phylogenetic diversity, functional diversity, nitrogen deposition, grassland ecosystem, global change

## Abstract

**Simple Summary:**

The continuous rise in anthropogenic nitrogen input, as one of the global change drivers, could have significant effect on terrestrial ecosystems, altering plant diversity and production. However, the effect of the nitrogen deposition rate on the multidimensional plant diversity–production relationship is poorly understood. Here, we investigated how varying rates of nitrogen deposition affect multidimensional plant diversity, biomass production, and its correlation in a temperate steppe of northern China. Biomass production increased initially and reached the maximum, then decreased with nitrogen deposition rates. Nitrogen deposition reduced species richness and plant functional diversity, while it enhanced plant functional trait identity such as plant height and leaf chlorophyll content. The phylogenetic structure of the plant community shifted from clustering to overdispersion along the nitrogen deposition gradient. However, nitrogen deposition did not change the relationships of species richness and phylogenetic structure with production but affected the functional diversity–production relationships. The results imply the robust species and phylogenetic diversity–production relationships and the varying functional diversity–production correlations under nitrogen deposition. The findings highlight the importance of a trait-based approach in studying the linkage between biodiversity and ecosystem function and facilitate the development of effective management strategies to maintain biodiversity and ecosystem function in the temperate steppe.

**Abstract:**

Nitrogen (N) deposition, as one of the global change drivers, can alter terrestrial plant diversity and ecosystem function. However, the response of the plant diversity–ecosystem function relationship to N deposition remains unclear. On one hand, in the previous studies, taxonomic diversity (i.e., species richness, SR) was solely considered the common metric of plant diversity, compared to other diversity metrics such as phylogenetic and functional diversity. On the other hand, most previous studies simulating N deposition only included two levels of control versus N enrichment. How various N deposition rates affect multidimensional plant diversity–ecosystem function relationships is poorly understood. Here, a field manipulative experiment with a N addition gradient (0, 1, 2, 4, 8, 16, 32, and 64 g N m^−2^ yr^−1^) was carried out to examine the effects of N addition rates on the relationships between plant diversity metrics (taxonomic, phylogenetic, and functional diversity) and ecosystem production in a temperate steppe. Production initially increased and reached the maximum value at the N addition rate of 47 g m^−2^ yr^−1^, then decreased along the N-addition gradient in the steppe. SR, functional diversity calculated using plant height (FDis-Height) and leaf chlorophyll content (FDis-Chlorophyll), and phylogenetic diversity (net relatedness index, NRI) were reduced, whereas community-weighted means of plant height (CWM_Height_) and leaf chlorophyll content (CWM_Chlorophyll_) were enhanced by N addition. N addition did not affect the relationships of SR, NRI, and FDis-Height with production but significantly affected the strength of the correlation between FDis-Chlorophyll, CWM_Height_, and CWM_Chlorophyll_ with biomass production across the eight levels of N addition. The findings indicate the robust relationships of taxonomic and phylogenetic diversity and production and the varying correlations between functional diversity and production under increased N deposition in the temperate steppe, highlighting the importance of a trait-based approach in studying the plant diversity–ecosystem function under global change scenarios.

## 1. Introduction

The interrelationship between plant diversity and ecosystem productivity is one of the most debated subjects in ecological research [1]. With rapid environmental changes, species loss is recognized as a threat to ecosystem function because species diversity has been documented as a major determinant of ecosystem productivity, stability, invasibility, and nutrient dynamics [2,3]. Recently, there has been a growing body of literature evidence that shows how diversity controls productivity [3,4]. However, the effect of environmental changes such as nitrogen (N) deposition on the diversity–productivity relationship is poorly understood [5]. 

Nitrogen is a key element controlling plant growth, survival, and many other biological processes, especially in N limited ecosystems [6]. Reactive N deposition from intensive agricultural, industrial, and other anthropogenic sources has been frequently recognized as a threat to global terrestrial biodiversity [6,7,8]. Global decline in biodiversity in turn will have direct negative impact on ecosystem functioning [2]. For instance, chronic N deposition induces the loss of species [9,10], homogenization of ecosystems, and alters the terrestrial carbon cycle [11]. Therefore, the relationship between plant diversity and productivity may be altered by N deposition in the terrestrial ecosystem. Some studies have found that N enrichment changes the strength or directions of the relationships of species richness (SR) with productivity and its stability [12]. In contrast, other studies have demonstrated that nutrient addition had no effect on the pattern of the SR-productivity relationships in grasslands [4] or an old field [13]. Apart from the studied ecosystem context, the level of N enrichment may be a dominant driver of the abovementioned contrasting results. However, the impact of simulating N deposition on the SR-productivity relationships is examined by comparing one level of N deposition simulation treatment with a control (i.e., N addition vs. without N addition) [8,14,15]. Few studies have been performed to test the SR-productivity relationship with varying levels of N deposition. In addition, most of previous studies about N deposition’s effect on the relationship between plant diversity and ecosystem productivity considered SR as the sole taxonomic diversity (TD) metric [5,10], neglecting how the underlying processes are mediated by different dimensions of plant diversity metrics such as phylogenetic (PD) and functional diversity (FD). Exploring phylogenetic community structure along the environmental gradient is vital for predicting ecosystem processes under global change [16]. Thus, studying the evolutionary relationships between plant species in natural communities under nutrient enrichment can reveal how global change drivers may impact different lineages and functional groups differently based on their shared evolutionary history and trait profile [16,17]. The trait-based approach is expected to explain species fitness, niche differences, and the community productivity response to global change [18]. Accordingly, integrating the phylogenetic and functional dimensions of biodiversity in the diversity–productivity relationship study can provide additional insights into biodiversity effects on ecosystem functioning rather than giving much attention to only one biodiversity dimension exclusively [17]. However, there are relatively few instances simultaneously assessing N deposition rates’ impacts on the various facets of biodiversity, which limits our comprehensive understanding of the effects of N deposition on the plant diversity–productivity relationships. Therefore, it is essential to examine the effect of N deposition rates on a comprehensive suite of diversity metrics beyond TD, which will enhance our ability to predict and gain a more complete understanding of the consequences of N deposition in the face of global change. 

Grasslands cover 40% of the land area in China [19]; however, nitrogen enrichment has negatively affected plant diversity and the ability of grasslands to absorb carbon [11]. The temperate steppe grassland of Inner Mongolia supports plant diversity, but it is highly sensitive to N deposition, overgrazing, and climate change [20]. A very limited number of studies have been undertaken on the impact of N deposition on plant diversity and productivity. Here, a manipulative field experiment with N addition rates was conducted in 2003 to explore how plant community structure and ecosystem function change along the N addition gradient in a temperate steppe of Inner Mongolia, Northern China. During the growing seasons of 2015 and 2016, plant community composition and biomass production were surveyed and plant functional traits such as plant maximum height and leaf chlorophyll content were measured in this study. TD, PD, and FD were calculated based on community investigation and functional traits measurement, and the mixed-effects model was employed to test N addition rates’ effects on the three dimensions of plant diversity. An analysis of covariance (ANCOVA) was used to examine the impact of N addition on the relationship between multidimensional plant diversity and production. The objectives of this study were to investigate the responses of TD, PD, and FD to N addition rates and to explore whether N addition rates affect multidimensional plant diversity–production relationships in the temperate steppe. 

## 2. Materials and Methods

### 2.1. Study Site

The study was conducted in a typical semiarid temperate steppe (116°17′ E, 42°02′ N) of Inner Mongolia in northern China. The study area is characterized by a temperate monsoon climate, having irregular rainfall and drastic changes in temperature. The mean annual temperature was 2.1 °C with the monthly mean temperature ranging from −17.5 °C in January to 18.9 °C in July. The amount of precipitation during the growing season (May–October) was 290.0 and 369.4 mm in 2015 and 2016, respectively. The soil at the study site is categorized as chestnut according to the Chinese soil classification system, and as Calsic-orthic Aridisol according to the US soil Taxonomy classification. The soil texture composition consisted of 62.75 ± 0.04% sand, 20.30 ± 0.01% silt, and 16.95 ± 0.01% clay [21].The soil available N content at a depth 0–10 cm was 0.19 mg g^−1^ and the pH value was 6.84 ± 0.07. The dominant grass species were *Stipa krilovii* and *Leymus chinensis* and the forb species were *Artemisia frigida* and *Potentilla acaulis* in the temperate steppe grassland [22]. 

### 2.2. Experimental Design

An extensive experimental research project, involving N addition and mowing treatments which simulated N deposition and grassland utilization, respectively, was launched to explore community structure and ecosystem function in 2003 in the temperate steppe of northern China. Eight blocks were established and each block consisted of eight plots which were randomly assigned to eight levels of N addition (N0, N1, N2, N4, N8, N16, N32, and N64 denote the N addition levels of 0, 1, 2, 4, 8, 16, 32, and 64 g N m^−2^ yr^−1^). Each plot had dimensions of 15 m × 10 m with a 4 m buffer zone between any two adjacent plots. Eight rates of N addition in the form of urea (CO(NH_2_)_2_) were added manually in each plot during mid-July from 2003. Among the eight blocks, four of them were mowed, and the clipped plants were moved out of the blocks at the end of each growing season since the project initiated. As part of the project, this study was performed to examine the effect of N addition rates on the relationships between multidimensional plant diversity and biomass production during two mid-growing seasons (June, July, and August) in 2015 and 2016 in the four unmowed blocks (each block consisted of 8 N addition plots, for a total of 32 plots). 

### 2.3. Community Survey and Plant Biomass Measurement

The community survey and biomass harvest were conducted during the middle of the local growing season (i.e., June, July, and August) in 2015 and 2016. A 50 cm × 50 cm quadrat was randomly placed in each plot for the vegetation survey, by placing the quadrats a minimum of 50 cm away from the edges of the plot to prevent any edge effects. All the vascular plants in each quadrat were identified to species level and recorded. The total coverage of individual plant species (%) in each sampling quadrat was estimated visually based on their occurrence. Based on cotyledon number and N fixation feature, all plant species were categorized into 3 different functional groups: forbs, grass, and legumes. Plant maximum height was calculated by averaging the height of three tallest individuals of each species in each quadrat, which was measured as height from ground level to the tip of the highest vegetative tissue using tapeline (cm). Leaf chlorophyll content was also measured by averaging the chlorophyll content of the same three individuals of each species using a Konica Minolta SPAD-502 Plus chlorophyll meter before harvesting plant biomass. The SPAD value typically ranges from 0 to 99, with higher values indicating a higher chlorophyll content. All aboveground plant tissues within the quadrat were clipped at the soil surface and sorted based on species identity before being put into different marked paper bags. Subsequently, belowground plant tissues and soil were sampled with a soil auger (20 cm in length, 5 cm in diameter). Two soil cores were collected and mixed for each quadrat. Plant roots and soil samples were separated from the mixed soil using a 2 mm sieve. Plant roots were cleaned with water and placed into marked paper bags. All plant samples were oven-dried at 70 °C for more than 48 h and weighed to determine AGB and BGB for each quadrat. AGB was estimated using the dry mass of all aboveground living plant tissues divided by the area of the quadrat, and BGB was evaluated with the dry mass of belowground plant tissues divided by the area of the two soil cores. The sum of AGB and BGB was used as a surrogate of ecosystem biomass production. 

### 2.4. Plant Diversity Metrics

We have examined three dimensions of biodiversity including TD, PD, and FD. TD included T0, T1, and T2 calculated using species-based Hill’s numbers (effective number of species) and represented species richness, the Shannon’s taxonomic diversity index, and the inverse Simpson diversity index, respectively, using the *hillR* package in R [23,24]. SR represents the number of species per plot. Community PD including the net relatedness index (NRI) and nearest taxon index (NTI) was estimated and a phylogenetic tree was made using the *Picante* and *ape* packages in R [25,26]. The phylogenetic tree generated from the dated molecular phylogeny of land plants was constructed by Zanne et al. [27]. The phylogenetic tree was extracted by using the *V.PhyloMaker* package in R [28]. The phylogenetic tree counted 40 tips, 1 for each species in the vegetation matrix, and had 39 internal nodes (see Appendix A). The NRI had a standardized effect size of the mean phylogenetic distance (MPD) of all species in the community [29]. The significance of NRI for an individual quadrat was assessed by comparing the observed MPD to a null distribution of MPD measured on 999 null communities. Thus, NRI is calculated as
$$NRI=-1\times \left({MPD}_{\mathrm{randomized}\;}-\;\mathrm{mean}\;\left({MPD}_{\mathrm{null}\;}\right)\right)/sd\left({MPD}_{\mathrm{randomized}}\right)$$
where MPD_randomized_ and sdMPD_randomized_ represent the mean and standard deviation of MPD estimated using 999 randomly sampled communities with given species diversity, respectively. Positive NRI values indicate that coexisting taxa are more related to each other than expected by chance (clustered dispersion) and negative NRI values indicate that coexisting taxa are more distantly related to each other than expected by chance (overdispersion) [29]. Similarly, NTI was calculated for each quadrat as alternate indicators of the plant community evolutionary relationships, by comparing the observed mean nearest taxon distance (MNTD) to a null distribution of MNTD measured on 999 null communities to see if they deviated significantly from random chance. NTI is calculated as
$$NTI=-1\times \left({MNTD}_{\mathrm{obs}\;}-\;\mathrm{mean}\;\left({MNTD}_{\mathrm{randomized}\;}\right)\right)sd\left({MNTD}_{\mathrm{randomized}\;}\right)$$

Two key functional traits were chosen based on the plant ecology strategy scheme because they are known to be functionally important and closely related to ecosystem functions (i.e., plant height and leaf chlorophyll content). These traits are also known to influence plants’ response to N addition [30]. The functional dispersion (FDis) of traits (the distribution of traits within a community) refers to the variation in functional trait values among species, which describes the mean distance in the trait space of individual species to the centroid of all species [31] calculated using both plant height and leaf chlorophyll content (i.e., FDis-Height and FDis-Chlorophyll content). In order to determine how plant functional traits responded to the experimental treatments at the community level, the community-weighted mean (CWM) value of each functional trait (functional composition of the community) was calculated using the plot level mean value of the trait weighted by the abundance of each species within the community. FD was computed using the *dbFD* function in *FD* package [32]. To avoid the analysis of highly correlated variables, we reduced the set of diversity metrics to SR, NRI, FDis-Height, FDis-Chlorophyll, CWM_Height_, and CWM_Chlorophyll_, which were strongly correlated to other diversity metrics (T1, T2, NTI, MPD, and MNTD) (Appendix A). Moreover, phylogenetic signals in functional traits (height and chlorophyll content) were tested following Pagel’s λ statistics [33] using the *phylosig* function in the *phytools* package to assess whether traits were phylogenetically conserved (i.e., to assess whether more closely related species have more similar functional traits) because the evolutionary relatedness of species may have an influence on the values of their traits. Thus, phylogenetic signals could indicate a tendency for closely related species to display similar trait values as a consequence of their phylogenetic proximity. 

### 2.5. Statistical Analysis

The Shapiro–Wilk normality test was employed to examine the normality of the data before statistical modeling. Biomass production was log transformed because the data showed strong positive skewness and to meet the assumptions of normality and heteroscedasticity. Correlations between diversity metrics and biomass production were assessed with Pearson’s correlation coefficients before modeling (Appendix A; Appendix A) to test the relationship between plant diversity and biomass production. We found strong correlations among the set of predictor variables (between SR and T1, T2, and PD) (Appendix A). Therefore, T1, T2, and PD were eliminated from further analysis. To explore the effect of N addition with time on the diversity metrics and biomass production, a linear mixed-effects model was used to assess the significance of the fixed effects using Type I analysis of variance (ANOVA) with N addition and time as fixed factors and plots set as the random factor. In the mixed-effects model, diversity metrics and biomass production were the response variables to N fertilizer over time and showed the interactive effect of N addition and time. The mixed-effect models were fitted using the *lmer* function in the *lme4* package [34]. Data were fitted to the models using the maximum likelihood to generate unbiased estimates of the model parameters. The homogeneity of variance among the means of response variables across treatment levels was visually examined using quantile–quantile plots of residuals for all models’ normality to assess whether model assumptions were met. A post hoc test (Tukey multiple comparison of treatment means across measuring times at a significance level of *p* < 0.05) was performed using the *emmeans* function in the *emmeans* package, to test the effect of various rates of N addition on plant diversity indices and biomass production. Moreover, regression analysis was employed to test the effect of the N addition rate on plant biomass production.

To test the effect of N addition on the pattern of the relationship between diversity metrics and biomass production, an analysis of covariance (ANCOVA) with the generalized additive model (GAM), generalized linear model (GLM), and generalized linear mixed model (GLMM) was implemented using the gam function in the *mgcv* package [35] and the *stats* and *glmer* function in the *lme4* package in R, respectively. In the GAM, GLM, and GLMM, N addition was treated as categorical factor and diversity indices were treated as covariates, while plot was included as a random factor. Both linear and quadratic forms of the covariates were included in the models. Diversity metrics in the GAM model were represented as a penalized regression spline (smoother) to detect nonlinear relationships, while N addition and measuring time were categorical factors. Optimal model selection was performed based on the lowest Akaike Information Criterion value (AIC: the lower the AIC value, the better the model). The final model was selected by simplifying the full GLM, GAM, and GLMM models in a stepwise deletion procedure using ΔAIC > 3 as the criterion for term deletion [36], following the procedure of Crawley [37]. GLM, GAM, and GLMM models were fitted with a Gaussian error distribution as the response was a continuous variable. We then used locally weighted regression smoothing with the *lowess* (scatterplot smoothing) function following model fitting to examine multidimensional plant diversity–biomass production relationships. All statistical analyses were performed using R 4.3.1. 

## 3. Results

### 3.1. Biomass Production along the N Addition Gradient

The range of production across all plots and measuring times was from 301.76 to 5985.30 g m^−2^ along the N addition gradient (Figure 1). The lowest production was observed in the N0 plot in June 2015, while the highest production was observed at the rate of 64 g N m^−2^ yr^−1^ in August 2016. When averaged for each treatment, production exhibited a nonlinear response pattern along the experimental N gradient (Figure 2). That is, biomass production initially increased and then decreased, with saturation thresholds appearing at approximately 47 g N m^−2^ yr^−1^. Maximum production (1909.91 g m−^2^) was calculated from the nonlinear regression at the N addition rate of 47 g N m^−2^ yr^−1^ in the temperate steppe (Figure 2). In this study, both the N addition and measuring time had a significant effect on production (both *p* < 0.001; Table 1). Compared with that in the N0 plots, production was increased by 74.9, 104.8, and 102.4% in the N16, N32, and N64 plots, respectively (all *p* < 0.01; Figure 1).

### 3.2. Multiple Plant Diversity along the N Addition Gradient

Species richness across all plots and measuring times varied from 1 to 17 along the N addition gradient in this study. The lowest SR occurred at the N64 plot in June 2015 and the highest SR was recorded at the N2 plot in August 2016. The significant proportional loss of SR relative to the control was 31.1, 43.3, 55.2, 64.1, and 73.7% under N4, N8, N16, N32, and N64, respectively (all *p* < 0.01; Figure 3a). NRI changed from positive (phylogenetic clustering) to negative (phylogenetic overdispersion) along the N addition gradient (Figure 3b). The relative change in NRI was −3.2 and −2.30 under N32 and N64 compared with that under N0, respectively (both *p* < 0.05 Figure 3b). The lowest FDis-Height and FDis-Chlorophyll (both 0.01) were observed in the N64 plots in August 2016 while the highest FDis-Height (1.95) and FDis-Chlorophyll (2.56) were observed in the N2 and N1 plots in July 2016, respectively (Figure 3c,d). FDis-Height was significantly decreased by 62.11 and 72.6% under N8 and N16, respectively (both *p* < 0.05; Figure 3c). FDis-Chlorophyll was reduced by 60.2 and 68.7% under N16 and N64, respectively (both *p* < 0.05; Figure 3d). The highest (63.30) and lowest (2.12) CWM_Height_ were observed under N64 in August 2016 and N0 in June 2015, respectively. The highest (55.13) and lowest (0.53) CWM_Chlorophyll_ occurred in the N64 plot in June 2015 and N0 plot in July 2016, respectively. CWM_Height_ and CWM_Chlorophyll_ were significantly affected by N addition and measuring time as well as the interaction between N addition and measuring time (all *p* < 0.05; Table 1). Compared with N0, CWM_Height_ was enhanced by 75.20 and 137.04% under N32 and N64, respectively (both *p* < 0.05; Figure 3e). CWM_Chlorophyll_ was stimulated by 28.1 and 111.6% under N16 and N64, respectively (both *p* < 0.05; Figure 3f). Phylogenetic signal Pagel’s λ values were 0.63 and 0.41 for height and chlorophyll content, respectively, suggesting that closely related species did not show similar trait values and the phylogenetic niche related to these traits was not conserved during evolution (Appendix A). Furthermore, based on the plants’ relative percent cover, *Artemisia_frigida, Potentilla acaulis*, and *Potentilla bifurca* were dominant forbs in the low N plots (i.e., in the control and 1 g N yr^−1^), whereas *Leymus chinensis* and *Stipa krylovii* were dominant grass species in the 32 and 64 g N yr^−1^ plots (Appendix A; Appendix A). 

### 3.3. Relationship between Plant Diversity and Biomass Production along the N Addition Gradient

The generalized additive model was retained for describing the impact of N addition on the relationships between plant diversity indices and biomass production based on the lowest AIC (Appendix A). The GAM model retained for interpretation, while the GLM and GLMM models were not retained for interpretation due to high model AIC (Appendix A and Appendix A). The GAM model revealed that N addition had no significant effect on the relationships of SR and NRI with production (Table 2). Both SR and NRI were negatively related to production across N addition rates in this study (Figure 4a,b). N addition did not affect the correlation between FDis−Height and production, but it significantly altered the relationships of FDis−Chlorophyll, CWM_Height_, and CWM_Chlorophyll_ with production (all *p* < 0.05, adjusted R^2^ = 0.53, 0.49, 0.64, respectively; Table 2). FDis−Chloropyll was negatively associated with production (*p* < 0.01; Figure 4d; Table 2), whereas CWM_Height_ and CWM_Chlorophyll_ exhibited positive effects on production (Figure 4e,f). 

## 4. Discussion

### 4.1. N Deposition Affects Biomass Production

The current study shows that biomass production initially increases along the N addition gradient until the optimum is reached and then slowly decreases with the N addition rate, indicating that the positive effect of N deposition on biomass production may diminish with a higher rate of N addition when N loading surpasses the N demand of plants. Previous studies have documented that after long-term chronic N inputs, N supply may exceed plant and microbial requirements, resulting in N saturation [9]. Numerous studies which only have two N addition rates (i.e., with N addition at the level of 6 or 10 g m^−2^ yr^−1^ and without N addition) have demonstrated that N addition increases biomass production [8,14,15]. However, in the present study under a low-to-moderate N addition rate (<47 g N m^−2^), biomass production increases, which might partly be attributed to the enhanced size of individual plants such as plant height and leaf chlorophyll content of dominant species (i.e., CWM plant height and leaf chlorophyll content) in line with previous studies [14,38]. This indicates that plant growth in this ecosystem was limited by nitrogen. Our study site in Inner Mongolia’s steppe soil had low N content (0.19 mg g^−1^), resulting in plant growth being limited by the low soil N content, as reported in a previous study [22]. Plant species that require higher nutrient levels could experience growth inhibition if they face prolonged nutrient deficiencies. This nutrient deficiency over time will limit the overall biomass production of those plant species [15]. Earlier studies showed that leaf chlorophyll content and plant height have a substantial impact on grassland productivity [39], as plant height significantly influences how plant communities respond to the addition of N by increasing light interception [30]. This could be attributed to the increased production of the dominant grasses offsetting the negative effects of species loss on biomass production. Hence, the dominant species, as controllers of ecosystem function, can provide short-term resistance to reductions in biomass production when species loss is nonrandom due to N deposition [6]. This indicates that N addition stimulates biomass production and promotes the turnover of species composition by favoring a few opportunistic species [40], which then neutralize the species loss effect on production. When the N enrichment rate approximately reaches 47 g N m^−2^ yr^−1^, biomass production reaches its maximum, indicating that N was a limiting factor at rates below 47 g N m^−2^ yr^−1^. It may be that N addition does lead to increased production but only up to a point where plants no longer efficiently utilize the added N. The reason for this pattern is likely due to the decrease in SR and FD because previous studies have shown that higher SR and variation in plant functional strategies increase biomass production [41,42]. Higher N addition rates promote competitive stress in the plant communities, as species have no similar competitive or stress-tolerant capability, as such, this result is partly in support of the stress gradient hypothesis, which suggests that the positive effect of biodiversity on ecosystem functioning can be enhanced when the environment experiences moderate levels of stress. However, if the stress levels increase beyond a certain point, the relationship between biodiversity and ecosystem functioning tends to diminish [43]. This indicates that traits related to resource acquisition, such as height and chlorophyll content, are no longer capable of offsetting the decline in biomass production resulting from the loss of species following N enrichment. 

In addition, changes in community composition as a result of high N supply may lead to the dominance of a small number of species, which may limit biomass production [44]. Meanwhile, under a higher rate of N addition (>47 g m^−2^ yr^−1^), biomass production declined, likely due to light competition and reduction in vegetation light penetration (the portion of light reaching the ground surface) [45], leading to species loss, specifically the elimination of highly productive forb species such as *Artemisia frigida* and *Potentilla acaulis* [20]. This shows that the negative effect of N addition on SR gradually dominates with increasing biomass production. The results suggest that higher N availability can offset the positive effects of diversity on biomass production. Nitrogen addition to grasslands increases the dominance of nitrogen-demanding grasses, which then suppress other slow-growing plant species, resulting in decreased productivity [6]. The results indicate that higher-rate N addition would be weaken commonly observed positive diversity–productivity relationships. In addition, the N−induced decline in biomass production at higher-rate N addition might be attributed to the loss of plant species due to the direct toxicity of N, depletion of soil nutrients like calcium, potassium, and phosphorus, lower soil pH and soil CEC, and increase in Mn ion toxicity [6,9]. Thus, this result suggests that competition and an increase in litter accumulation can reduce seedling establishment following a higher rate of N addition leading to light and water limitation [9], resulting in a greater reduction in plant diversity and plant biomass [40]. The fact that N enrichment may make the water limitation in temperate steppe grasslands worse [44] may also be a contributing factor to the decreased biomass production following high rates of N addition. The increased biomass production resulting from N addition can lead to higher plant water demand for transpiration and metabolism [42]. Consequently, our findings confirm that a low rate of N addition induces a trade-off between plant diversity and biomass production, but higher rates of N addition decrease both plant diversity and biomass production in a temperate steppe of Inner Mongolia. This finding is consistent with research studies reporting that diversity and productivity are interdependent [3]: resource supply drives diversity, and diversity drives resource usage, which then influence biomass production simultaneously. In addition, the contrasting effect of N deposition on FD and biomass production, with the former decreasing and the latter increasing with an increasing N rate, might be attributable to intraspecific trait variation promoted by N addition, which was observed in previous studies [38,46]. Given the positive effect of N addition on CWM_Height_ and CWM_Chlorophyll_ (community functional trait composition) and the negative effect of N addition on FDis-Height and FDis-Chlorophyll content with biomass production, our results likely reflect that N addition promotes the growth of highly productive species at the expense of short-statured and slow-growing species. Consequently, the biomass production under multiple N additions in a temperate steppe is determined by the functionally dominant species that attained the largest biomass (i.e., dominance effect). This is consistent with the results of the positive links between CWM of plant traits and biomass production under nutrient enrichment, emphasizing that the functional traits related to resource acquisition (plant height and leaf chlorophyll content) may play a significant role in response to N addition, driving the plant diversity−biomass production relationship in the temperate steppe. Thus, we found support for the mass ratio hypothesis [47], which indicates that biomass production is primarily determined by the functional traits of the dominant species instead of variation in ecological strategies, reflecting from the fact that a community composed of species with a high trait value could increase biomass production. That is, the high biomass production under low N addition rates would depend on the functional traits of the dominant grass species [48]. Tatarko et al. [38] also observed that dominant species’ traits can play a significant role in driving grassland primary biomass production under N enrichment. In part accordance with our expectations, this may be a result of intense interspecific competition following N addition, indicating that the addition of N shifted few species toward extreme trait values [48]. Therefore, this finding suggests that, to reduce light competition resulting from N enrichment and improve biomass production in the grassland community of inner Mongolia, moderate mowing or grazing that can remove or open up the litter layer is vital to enhance seedling establishment and maintain biodiversity and production, particularly in the plant communities with abundant litter.

### 4.2. Effect of N Addition Rate on Multidimensional Plant Diversity

The present study supports previous findings showing that species richness decreases under N addition [46,49]. The decline in species richness may be attributed to size asymmetry light competition and changes in underground processes, such as soil acidity and the direct toxicity of N, which impairs electron transport in chloroplasts [6,9,45]. Additionally, this decrease may also be partially driven by nutrient imbalance in plants [6]. In general, according to a global meta-analysis [50], species loss induced by elevated N might be explained by two aspects. On one hand, the accumulation of a large amount of standing litter and higher live biomass production increases size asymmetry competition for light [45,51]. Previous nutrient addition tests have demonstrated that under high N supplies, plants’ competition shifts from soil N to light [20,52], which may also be a factor in the loss of species along the N gradient by decreasing the proportion of photosynthetically active radiation transmitted through the canopy to the ground surface [40]. On the other hand, N deposition alters the soil chemistry, resulting in the limitation of basic cations such as calcium, potassium, and phosphorus in the soil [30], which can cause soil acidification and ammonium toxicity when the magnitude of N input surpasses the plants’ ability to utilize it [7,53], thereby reducing SR [10,49] and altering the composition of the soil microbial community [9]. In the Inner Mongolia temperate steppe, it has been reported that N addition decreased forb species due to the accumulation of inorganic N, a decrease in soil pH, and an increase in toxic metallic ion concentrations such as Mn and Al, resulting in the depletion of basic cations such as Ca, Mg, and K [54]. The increase in CWM_Height_ along the gradient of N addition may have stimulated rising competition for light among coexisting species. Species that are short-statured, leguminous, locally rare, and capable of tolerating low nutrient levels are likely to be especially susceptible to loss due to competitive exclusion following N-induced increases in biomass production [8,55]. Meanwhile, our results demonstrated a considerable variation in the response pattern of plants from different functional groups following varying rates of N addition, which was consistent with previous findings [22,56]. The higher rate of N addition (32 g m^−2^ N yr^−2^ and above) leading to the loss of dominant forbs such as *Artemisia frigida* and *Potentilla acaulis* (on average, made up in the order of 35% and 20% of the relative percent cover, respectively) in the control (N0) and N1 (1 g N yr^−1^) plots, respectively. In the higher-rate treatments (i.e., over 16 g N yr^−1^), perennial rhizomatous grasses such as *Leymus chinensis* (a perennial rhizomatous clonal grass; on average, comprises 65% relative percent cover) dominated, which to a large extent supported the previous results [20]. This indicates that there is variability in the responses of different functional groups of plants to nutrient addition. This finding is consistent with previous studies, which have found that forb species in a temperate steppe of Inner Mongolia grassland are more sensitive to N−induced stress, whereas grass such as *Leymus chinensis* is positively associated with N addition [44,54]. Grass could have a higher nitrogen utilization efficiency compared to forbs [56]. As such, grass species outcompete forbs by growing at a faster rate and exploiting light more efficiently, which hinders the growth of forbs [45]. Similarly, Song et al. [20] observed that higher rates of N addition promote the dominance of grasses, which outcompete forbs in a temperate steppe ecosystem. Although, a study conducted by DeMalach et al. [45] demonstrated a decrease in forb diversity following N enrichment in the Mediterranean grassland. This result suggests that N−induced changes in plant diversity are species-specific, possibly due to the difference in the sensitivity of species to nutrient addition and resource utilization strategies [38], and in turn, fast-growing and short-lived forb species are more sensitive to N deposition [19]. This might be related to, at least in part, the relatively large rhizosheath size on grass species providing resistance against N induced stress compared to forbs. More recently, it has been reported that rhizosheath size plays a substantial role in grass relative to forbs in providing resistance against N induced metal stress by releasing protective compounds (such as organic acids and mucilaginous compounds), which in turn regulate the competitive advantage for grass at the cost of forbs by colonizing the belowground habitat [49]. In addition, clonal plants (such as *L. chinensis* in our experiment) have competitive advantages over forbs in acquiring soil nutrients and light, due to clones and greater root biomass [54]. Thus, N regulates the growth of grass to become taller and possess a dominant canopy. This allows the grass to intercept available light and increase resource use efficiency such as light. This result provides support for the light asymmetry hypothesis, which posits that tall individuals receive a higher amount of light per unit biomass than shorter individuals [45], which, in turn, results in the loss of forb species in grasslands [30,56]. Another important process, beside light competition, that can likely lead to species loss related to N addition is that high levels of soil resources reduce the niche dimension [40,44], which in turn favors a few dominant plant species by affecting competitive balance among species owing to a reduction in the number of limiting resources. Our study in a temperate steppe of Inner Mongolia revealed loss of species along the N addition gradient probably resulting from both light competition and N-induced change in soil chemical properties [49]. 

In the present study, the shifting of phylogenetic diversity from phylogenetic clustering to overdispersion (i.e., high phylogenetic dissimilarity) owing to elevated N fertilization is consistent with previous findings in a semi-natural plant community [57]. However, given that closely related species are more likely to share similar traits (i.e., have a high phylogenetic signal), our findings are in contrast to the prediction that closely related species may exhibit similar sensitivity or resistance to a particular perturbation [58]. However, our findings are congruent with the theory that more closely related species are less likely to coexist, which means that closely related species compete more intensely for resources such as light [29], resulting in phylogenetic overdispersion [16]. A possible reason is that some functionally important and productive clades outcompeted other short-statured groups resulting in species loss under N enrichment, which may lead to phylogenetic dissimilarity among coexisting plants. This suggests that plant species that are resilient to stress caused by N are not limited to a certain plant family and may have evolved separately in different lineages [57]. Thus, competition might be one possible reason and our work provides evidence for this phenomenon. N addition may act as an environmental filter on ecologically similar non-related species, which might cause phylogenetic overdispersion (i.e., convergent evolution) by filtering lineages based on their functional traits [59]. This suggests that light competition among coexisting species under N enrichment and N toxicity might be a potential cause of phylogenetic overdispersion, as it increases the likelihood of extinction of short-statured and rare species while promoting the coexistence of multiple distinct lineages [16]. Consequently, N−induced competition among closely related species leads to favoring the coexistence of distantly related species with distinct ecological strategies [57]. Plant height and chlorophyll content were used in the current investigation but failed to capture a strong phylogenetic signal, indicating that these traits may have experienced phylogenetically convergent evolution throughout the course of evolutionary time [58]. This suggests the possibility that the management strategies to conserve plant diversity in the temperate steppe of Inner Mongolia, such as sowing seeds of phylogenetically distantly related species (particularly grass species), could be crucial to maintain plant diversity in the face of N deposition.

Functional diversity decreased along the N addition gradient in the current study, indicating that the nutrient gradient in grassland affects the functional structure of communities [59]. For instance, Xu et al. [41] observed that biotic processes such as competition can produce trait divergence by excluding similar species (i.e., promoting niche difference). This implies that dominant plants tend to eliminate neighboring short-statured plants that have similar traits. N addition may lead to the emergence of new environmental conditions that favor the selection of new traits, while disadvantaging the common species and accelerating rates of competitive exclusion [45,55]. In turn, nutrient addition can promote the dominance of species with similar functional traits that have less overlap in their use of resources and facilitate the coexistence of species that share similar traits, leading to a more homogeneous plant community. One possible reason for this phenomenon is that N-induced competition for light affects plant functional traits that are related to light acquisition, such as the height of dominant species. This increase in height then leads to a reduction in functional diversity [51], by causing the elimination of short-statured species. Consequently, the overall FD of the plant community becomes low. As such, the potential factor for the loss of species might be linked to the differences in resource acquisition and resource storage strategies among species (grass vs. forbs) [44]. These results are similar to those of other studies [19,57], which indicate that N fertilization enhances competitive response traits such as CWM_Height_, which may accelerate light competition intensification among coexisting plant species. Plant height is a key functional trait that affects the competitive ability for light. Plants with a short stature and resource storage strategy particularly suffer more from reduced light availability [9]. However, our results contradicted some other previous findings [60] that N addition had no direct and significant effect on FD and PD in alpine meadow and alpine steppe grassland. The discrepancy may be related to many reasons, including the difference in annual mean temperature, plant species composition, different forms of N added, and soil properties between alpine grasslands and temperate steppe [42]. The mean annual temperature is higher in the temperate steppe compared to the alpine meadow and alpine steppe, as higher temperature promotes the uptake of soil N by plants [55]. Moreover, N addition increases CWM_Chlorophyll_, which is an important pigment for photosynthesis [39]. For instance, a previous study found that N addition increases CWM_Chlorophyll_ by 34% (i.e., 111 mg cm^−2^) [38], compared with the control. Through this understanding, the results further confirmed that the dominating species with a resource-acquisitive strategy was a mediator of the effects of N on plant diversity [53]. These results suggest that nutrient availability increase plant strategy towards resource acquisition [38]. Therefore, future management that focuses on locally susceptible species to N-induced stress, particularly short-statured forbs, will be vital to maintain the highest levels of biodiversity in the temperate steppe.

### 4.3. Effect of N Addition Rate on the Relationship between Multidimensional Plant Diversity and Biomass Production

Our results have demonstrated that N addition did not significantly alter the strength or direction of the relationship of SR, NRI, and FDis-Height with biomass production, despite the fact that N addition significantly decreases the aforementioned diversity indices. It is observed that there is an inverse relationship between the increase in biomass production caused by N addition and the decline in plant diversity. In other words, as the biomass production increases, the plant diversity decreases, and vice versa following N enrichment, which agrees with the results of previous global syntheses [4,52]. The direction and strength of the relationship between SR and biomass production in response to nutrient addition varies between contexts, systems, and sites [40]. The present study revealed that the plant community tends to be phylogenetically overdispersed at higher N addition rates, implying that species are distantly related, but N addition did not change the negative correlation between NRI and biomass production. A possible reason could be that elevated N may exacerbate intense light competition between closely related species, which may have increased the likelihood of the coexistence of distantly related species that are more tolerant of elevated N. The functional dispersion of leaf Chlorophyll content exhibited a negative relationship with biomass production under the N addition gradient. This may result from the N-induced changes in functional diversity, which decrease resource use efficiency such as light [51], presumably reflecting the effect of functional filtering, which in turn reduces biomass production by inhibiting the diversity of traits and decreasing the resource use efficiencies owing to declining diverse nutrient-uptake strategies during the different periods of growing [4,30]. Our findings showed that functional trait diversity in response to N enrichment has a negative relationship with biomass production. This indicates that the deposition of N leads to a decrease in trait spaces, which in turn influences the link between functional diversity and biomass production in the temperate steppe grassland. Plant communities have species which possess distinct traits, utilize diverse resources, and maintain biomass production [53], but the enrichment of varying rates of N promotes the growth of plant species that have similar functional strategies [59]. However, these species may have low efficiency in utilizing resources, which leads to a decrease in biomass production [30]. In accordance, numerous research studies have demonstrated that in long-term grassland experiments, nitrogen enrichment usually increases grassland productivity while decreasing plant diversity [5,15,51], suggesting that the generally observed positive correlation between diversity and biomass production would be diminished with N addition [40,52]. Taken together, our findings in a temperate steppe of Inner Mongolia demonstrate that elevated N alters the composition and structure of the community. Consequently, this could counteract the positive correlation between plant diversity and biomass production (i.e., biomass production was influenced by community structure), consistent with prior investigations [20,44]. Our findings indicate that the addition of nitrogen decreases SR, NRI, and FDis-Height, and biomass production was also negatively associated with the aforementioned facets of diversity metrics. However, the CWMs of plant height and leaf chlorophyll content were significantly positively related to biomass production under N addition. The results suggest the dampening of positive interactions among the grassland species and an increasing contribution of a few dominant species to a higher biomass production in response to increasing N accumulation. Measuring only two functional traits may have been a limitation of our study because the effect of functional diversity on the relationship between plant diversity and biomass production depends on which traits are included in the analysis [53]. Moreover, we only conducted this experiment for two years. For a comprehensive understanding of the impact of N addition on biodiversity–productivity relationships, a longer time series of data is required. The response of the plant diversity–production relationship to N deposition varies with the rate and duration of N addition [7], and these limitations may introduce bias into the results due to the relatively short duration of the study.

## 5. Conclusions

In this study, N deposition significantly altered multidimensional plant diversity and ecosystem production in the temperate steppe. Forbs exhibited more sensitive responses to N deposition-induced stress than grass, leading to a decline in SR along the N deposition gradient. The phylogenetic structure changed from clustering to overdispersion with N deposition rates by filtering out lineages with similar functional traits. Plant height and leaf chlorophyll content-based functional dispersion reduced, but CWM_Hight_ and CWM_Chlorophyll_ increased with N deposition rates, implying that N deposition−induced stress favored tall species with high leaf chlorophyll content in the temperate steppe. Ecosystem production increased, reached the maximum, and then decreased with N deposition rates. However, N deposition had no impact on the relationships of SR and PD with production, but it affected the relationships of FDis-Chlorophyll, CWM_Height_, and CWM_Chlorophyll_ with production. The findings highlight the importance of a trait-based approach for understanding the multidimensional biodiversity–ecosystem function in grassland ecosystems under N deposition scenarios. For a better understanding of the N deposition effect on ecosystem structure and function, long-term manipulative experiments incorporating multiple functional traits, environmental factors, and soil microbial analysis are desirable. Based on the findings, to mitigate N−induced species loss and improve ecosystem production in the grassland of Inner Mongolia, moderate mowing or grazing may be desirable to facilitate seedling establishment and maintain plant diversity. Additionally, it is recommended to establish a critical limit for N deposition in order to implement the idea of critical loads. This limit would help to restrict the amount of N load and mitigate its subsequent negative effects on both biodiversity and ecosystem production.

## Figures and Tables

**Figure 1 biology-13-00554-f001:**
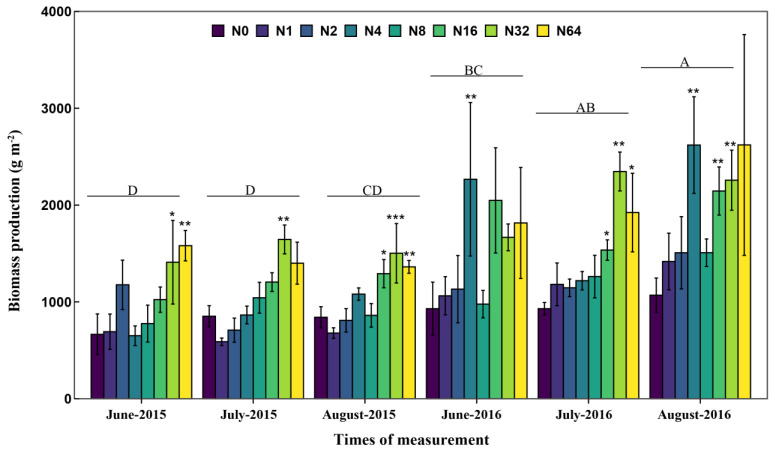
Effect of N addition rate on plant biomass production (mean ± SE) during the growing seasons of 2015 and 2016 in a temperate steppe of Inner Mongolia, northern China (*n* = 192; Error bars represent the standard error of the mean). N0, N1, N2, N4, N8, N16, N32, and N64 represent nitrogen fertilizer rates of 0, 1, 2, 4, 8, 16, 32, and 64 g N m^−2^ yr^−1^, respectively. Asterisks indicate significant differences between the treatment and control groups * *p* < 0.05, ** *p* < 0.01, *** *p* < 0.001. Different letters indicate significant differences between measuring times (*p* < 0.05).

**Figure 2 biology-13-00554-f002:**
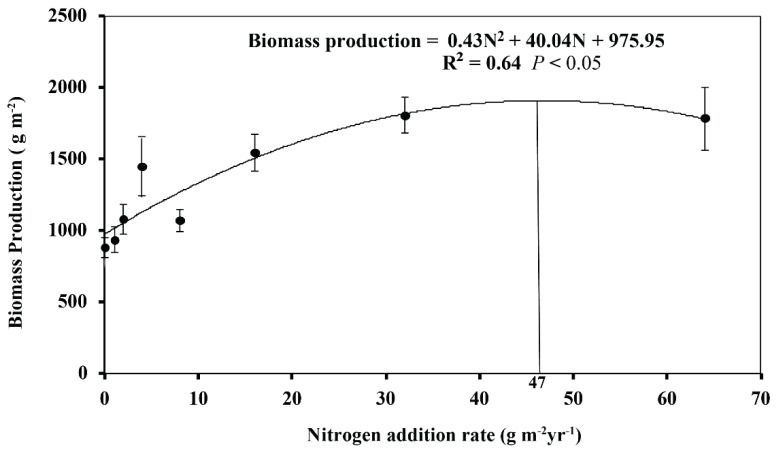
Response of biomass production to N addition rate in a temperate steppe of Inner Mongolia, northern China. Maximum biomass production (i.e., 1909.91 g m^−2^) was calculated from the second-degree quadratic regression at the rate of 47 g N m^−2^ yr^−1^. Each data point represents the mean (±SE) biomass production across all measuring times. The vertical line represents the point where maximum biomass production was calculated from second-degree quadratic regression. Error bars show the mean ± 95% confidence intervals.

**Figure 3 biology-13-00554-f003:**
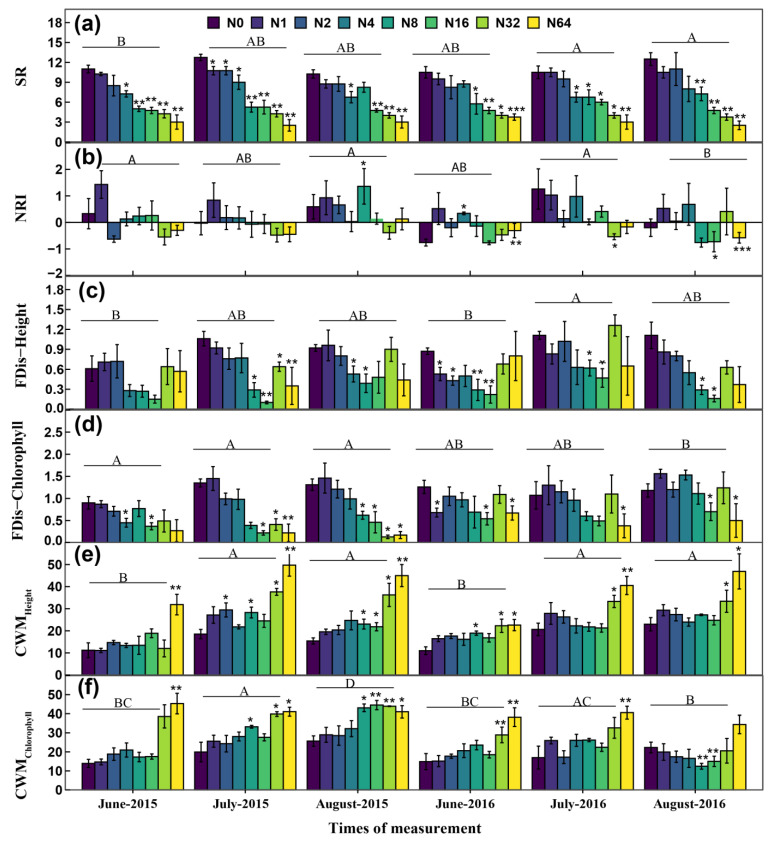
Effect of N addition gradient on multidimensional plant diversity metrics (mean ± SE) (Tukey pairwise tests on the difference of plant diversity indices) including species richness (SR; (**a**)), net relatedness index (NRI; (**b**)), functional diversity of plant height (FDis−Height; (**c**), functional diversity of chlorophyll content (FDis−Chlorophyll; (**d**)), community−weighted mean of plant height (CWM_Height_ (cm); (**e**)), and community−weighted mean of chlorophyll content (CWM_chlorophyll_; (**f**)) in a temperate steppe of Inner Mongolia, northern China. N0, N1, N2, N4, N8, N16, N32, and N64 represent nitrogen fertilizer rates of 0, 1, 2, 4, 8, 16, 32, and 64 g N m^−2^ yr^−1^, respectively. Error bars represent standard error of the mean (*n* = 192). Asterisks indicate significant differences between the treatment and control groups * *p* < 0.05, ** *p* < 0.01, *** *p* < 0.001. Different letters indicate significant differences between measuring times (*p* < 0.05).

**Figure 4 biology-13-00554-f004:**
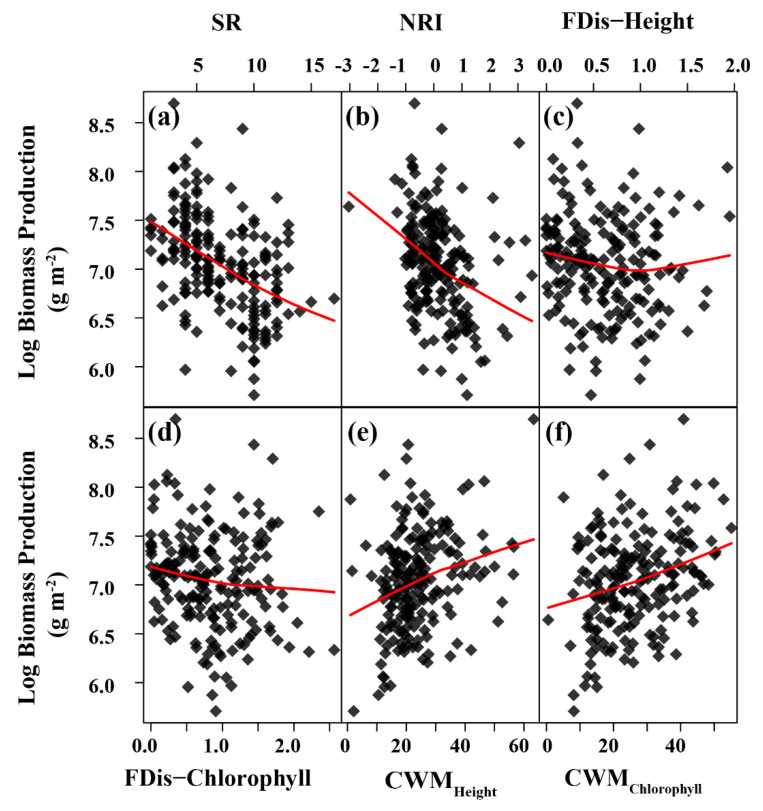
Relationships between biomass production and plant diversity indices including species richness (SR; (**a**)), net relatedness index (NRI; (**b**)), functional diversity of plant height (FDis−Height; (**c**)), functional diversity of chlorophyll content (FDis-Chlorophyll; (**d**)), community−weighted mean of plant height (CWM_Height_; (**e**)), and community−weighted mean of leaf chlorophyll content (CWM_Chlorophyll_; (**f**)) across eight levels of nitrogen (N) addition during the growing season in a temperate steppe of Inner Mongolia, northern China. Solid lines are fitted Lowess-smoothed curves. Each data point represents the measured value of individual plots for each measuring time (*n* = 192).

**Table 1 biology-13-00554-t001:** Summary (F-values) of mixed−effects model for the effects of nitrogen (N), time, and their interaction on plant diversity indices (species richness, SR; net relatedness index, NRI; functional diversity of plant height, FDis−Height; functional diversity of chlorophyll content, FDis−Chlorophyll; community−weighted mean of plant height, CWM_Height_; and community−weighted mean of chlorophyll content, CWM_Chlorophyll_) and biomass production in a temperate steppe of Inner Mongolia, northern China.

Source	df	SR	NRI	CWM_Height_	CWM_Chlorophyll_	FDis-Height	FDis-Chlorophyll	Production
N	7	32.79 ***	4.08 **	11.66 ***	15.38 ***	4.49 **	5.36 ***	8.66 ***
Time	5	1.45	3.04 *	44.23 ***	24.17 ***	4.18 **	7.70 ***	15.56 ***
N × Time	35	1.06	1.07	2.11 **	1.74 *	0.87	1.78 *	0.97

Note: Statistical significance is indicated as * *p* < 0.05, ** *p* < 0.01, *** *p* < 0.001, df, degree of freedom.

**Table 2 biology-13-00554-t002:** Analysis of deviance results for the generalized additive model (GAM) testing the responses of biomass production to species richness (SR), net relatedness index (NRI), functional diversity of plant height (FDis−Height), functional diversity of chlorophyll content (FDis−Chlorophyll), community−weighted mean of plant height (CWM_Height_), and community−weighted mean of chlorophyll content (CWM_Chlorophyll_) across eight levels of nitrogen (N) addition in a temperate steppe of Inner Mongolia, northern China. CWM_Height_ × N: interactive effects of CWM_Height_ and N addition; qFDis−Height and qCWM_Height_: second degree of the quadratic term of FDis−Height and CWM_Height_, respectively; CWM_Height_ × N: interactive effects of CWM_Height_ and nitrogen; CWM_Chlorophyll_ × Time: interactive effects of CWM_Chlorophyll_ and measuring times; N: various rates of nitrogen fertilizer addition; Time: measuring time (i.e., repeated measures in June, July, and August of 2015 and 2016); N × Time: interactive effects of N and time retained in the model.

Response	Term	Statistics			
Biomass Production	Taxonomic diversity	df	F	*p*	AIC
	s(SR)	1	0.95	0.32	177.22
	s(qSR)	1	2.27	0.13	
	N	7	2.49	**<0.05**	
	Time	5	16.27	**<0.001**	
	Phylogenetic pattern	df	F	*p*	AIC
	s(NRI)	1	0.27	0.61	179.17
	N	7	7.74	**<0.001**	
	Time	5	15.15	**<0.001**	
	Functional Diversity	df	F	*p*	
	s(FDis-Height)	1	1.68	0.90	178.80
	s(qFDis-Height)	1	1.75	0.18	
	N	7	7.10	**<0.001**	
	Time	5	15.72	**<0.001**	
	s(FDis-Chlorophyll)	1	5.26	**<0.05**	170.20
	FDis-Chlorophyll × N	7	3.28	**<0.01**	
	N	7	4.72	**<0.001**	
	Time	5	15.00	**<0.001**	
	Functional composition	df	F	*p*	AIC
	s(CWM_Height_)	1	5.17	**0.05**	183.57
	s(qCWM_Height_)	1	4.84	**<0.05**	
	CWM_Height_ × N	7	2.50	**<0.05**	
	qCWM_Height_ × N	7	2.27	**<0.05**	
	N	7	3.21	**<0.01**	
	Time	5	14.63	**<0.001**	
	s(CWM_Chlorophyll_)	1	32.65	**<0.001**	141.64
	CWM_Chlorophyll_ × N	5	8.35	**<0.001**	
	N	7	2.22	**<0.05**	
	Time	5	9.32	**<0.001**	
	N × Time	35	2.33	**<0.001**	

Significant effects (*p* < 0.05) are indicated in bold.

## Data Availability

The datasets analyzed during the current study are available from the corresponding author on reasonable request.

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
