# Peer review of "Effect of Nitrogen Application Rate on the Relationships between Multidimensional Plant Diversity and Ecosystem Production in a Temperate Steppe"

_biology, 2024, doi:10.3390/biology13080554_

Round 1

Reviewer 1 Report

Comments and Suggestions for Authors

The approach and research work carried out in this work are of great ecological interest, and I personally want to recognize the great effort that the researchers made in carrying out this process since 2003. However, there are some issues:

1. The last paragraph (from 112 line to 121) of the introduction should be reconsidered; the authors do not analyze competition for light in their experiments. This paragraph should be written to better relate what they expected, what they analyzed, and the result.

2. I would like to know why the authors did not analyze the nitrogen content before conducting their experiments. If you have this data, you should place it in materials and methods where the general characteristics of the study site are described and then return to it in the discussion, where you mention that the positive effect of nitrogen on biomass accumulation could be due to the fact that the study site had low levels.

3. According to what was described in your materials and methods, you categorized plant species on 3 different functional groups: Forbs, Grass and Legumes. The results never talk about these three groups. What is the most abundant group according to nitrogen treatments? Describing it in an additional figure would be the most appropriate.

4. Are tables 1 and 2 the statistical analyzes corresponding to figures 1 and 2? Statistical analysis should be indicated with asterisks or letters in the graphs, so figures should be modified and tables removed.

5. Compare numerically the amounts of nitrogen used by other authors. It is mentioned that the other authors use two treatments “without nitrogen” and “with nitrogen” (How much?). Line 391.

6. The authors should be more forceful when discussing their results. I would like to know what your answer is to the following questions: What is your conclusion about N deposition? Is it good or bad for plant communities?

5. Check the double spaces between words.

Author Response

Reviewer 1

The approach and research work carried out in this work are of great ecological interest, and I personally want to recognize the great effort that the researchers made in carrying out this process since 2003. However, there are some issues:

  1. The last paragraph (from 112 line to 121) of the introduction should be reconsidered; the authors do not analyze competition for light in their experiments. This paragraph should be written to better relate what they expected, what they analyzed, and the result.

Author’s responses: Thank you for your valuable comments and suggestions. The last paragraph of the Introduction was reconsidered and rewritten as suggested in the revision (Page 3, Lines 127-135). As suggested, we first deleted the statement about competition for light, which we did not analyze in this study, and then added the multidimensional plant diversity indices we calculated, the methods used in data analysis, and the objectives we aimed for in the revised version.

  1. I would like to know why the authors did not analyze the nitrogen content before conducting their experiments. If you have this data, you should place it in materials and methods where the general characteristics of the study site are described and then return to it in the discussion, where you mention that the positive effect of nitrogen on biomass accumulation could be due to the fact that the study site had low levels.

Author’s responses: Thank you for the valuable suggestion. We had the data of the soil textural composition, soil nitrogen content, and soil pH at study site before conducting the experiment, and placed them in the study site description of the Materials and Methods (Page 4, Lines 144-149). Then, in the Discussion section we explained why biomass production increased in response to N addition in the steppe. That is, N addition could mitigate the nutrient limitation of plant growth leading to enhanced biomass production in the temperate steppe with low N content (Page 14-15, Lines 427-432).

  1. According to what was described in your materials and methods, you categorized plant species on 3 different functional groups: Forbs, Grass and Legumes. The results never talk about these three groups. What is the most abundant group according to nitrogen treatments? Describing it in an additional figure would be the most appropriate.

Author’s responses: Thanks for your constructive comments. All species occurred in the field survey was classified into 3 groups according to the number of cotyledon and the feature of nitrogen fixation. We made this point clear in the M & M of the revised manuscript (Page 4, Lines 178-180). Most dominant species based on percent relative coverage along nitrogen application rates were shown in an additional figure (Fig. S4) and the functional groups of the dominant plant species were also listed in an additional table (Table S1) (Supplementary materials). Based on Fig. S4 and Table S1, we showed the dominant species and group under low and high nitrogen addition, respectively, in the Results section of the revision (Page 11, Lines 365-368). Also, we explained the responses of plant species from different functional groups to N addition rates in the Discussion section (Page 17, Lines 537-548).

  1. Are tables 1 and 2 the statistical analyzes corresponding to figures 1 and 2? Statistical analysis should be indicated with asterisks or letters in the graphs, so figures should be modified and tables removed.

Author’s responses: Yes, Tables 1 and 2 are the statistical analysis corresponding to Figures 1 and 3. We have updated Figures 1 and 3 indicating the statistical results with asterisks and letters, and Table 2 removed in the revision (Page 8, Lines 319-326 and Page 12, Lines 371-381). However, we kept Table 1 in the revision as it showing the main results of repeated measures mixed effect model about the effects of nitrogen addition and measuring time on plant diversity metrics and production, and provided some statistical information (such as degree of freedom, statistics values, interaction effects) which were not included in the figures and main text. So, Table 1 was left in the revised manuscript.

  1. Compare numerically the amounts of nitrogen used by other authors. It is mentioned that the other authors use two treatments “without nitrogen” and “with nitrogen” (How much?). Line 391.

Author’s responses: Thanks for pointing this issue out. Most previous studies have only two levels of N addition (with vs. without N addition) to test N addition impact on plant diversity and production. However, the amount of N addition used in previous studies is often different, mostly ranging from 2 to 20 g N m-2 yr-1. In the revised manuscript, we added the amount of N addition from three previous studies we cited in this sentence (Page 14, Line 421-422).

  1. The authors should be more forceful when discussing their results. I would like to know what your answer is to the following questions: What is your conclusion about N deposition? Is it good or bad for plant communities?

Author’s responses: Thank you for the valuable comment. From the production point of view, a low to moderate rate of N deposition (< 47 g N m-2 yr-1) stimulates biomass production, but a high rate of N deposition (> 47 g N m-2 yr-1) reduces biomass production in the temperate steppe. From the biodiversity conservation point of view, species diversity decreases along the N deposition gradient, whereas the phylogenetic structure of plant community changes from clustering to overdispersion, implying that distantly related species occurs at a high rate of N deposition. In addition, N deposition favors tall species with high chlorophyll content, resulting in decreased functional dispersion and increased CWM values. Our results showed a trade-off between plant diversity and biomass production under N deposition, which is a great challenge for the future sustainable management of the temperate steppe. We updated the conclusion and made it more focus and clear in the revision (Page 20, Lines 697-720).

  1. Check the double spaces between words.

Author’s responses: Thanks a lot. Done.

Reviewer 2 Report

Comments and Suggestions for Authors

Please find attached Ms. pfg file for suggestions and edits. 

Comments on the Quality of English Language

English quality is acceptable, some minor improvements can be applied. 

Author Response

Please see the attached response file.

Reviewer 3 Report

Comments and Suggestions for Authors

Interesting manuscript. The figures need formatting. It would have been great if there was a separate conclusion section that would help to conclude the overall manuscript addressing the two questions mentioned in the introduction. What are the possible strategies for reducing the negative impacts of nitrogen addition on plant diversity while promoting biomass production? could be included. Is there any seasonal variation in the effects of N addition or deposition on plant biodiversity/community? Are there any effects of Nitrogen addition or deposition on pollination process? A brief future directions in such kind of manuscript would be nice. Overall, a good manuscript. 

Author Response

Reviewer 3

Comments and Suggestions for Authors

Interesting manuscript. The figures need formatting. It would have been great if there was a separate conclusion section that would help to conclude the overall manuscript addressing the two questions mentioned in the introduction. What are the possible strategies for reducing the negative impacts of nitrogen addition on plant diversity while promoting biomass production? could be included. Is there any seasonal variation in the effects of N addition or deposition on plant biodiversity/community? Are there any effects of Nitrogen addition or deposition on pollination process? A brief future direction in such kind of manuscript would be nice. Overall, a good manuscript. 

Author’s responses: Thank you for your positive comments. As suggested, we updated figures 1 and 2 in the revision. Also, we made a separate conclusion section including possible strategies (such as moderate mowing and grazing) for reducing the negative impact of N addition on plant diversity and a brief future direction (establishing N deposition critical level) based on our results in this research field at the end of the main text (Page 22, Lines 703-726).

In this study, we showed the significant seasonal variation in the effects of N addition on plant diversity metrics and production (measuring time effect; Table 1), and mentioned the measuring time effect in the Results (Page 7, Lines 310-311). We did not examine the effect of N addition on pollination in this study, although pollination change under N addition may contribute to the variation in community composition and plant diversity. Thanks for your interesting question. We will consider in our future research work to explore the effect of N deposition on plant pollination process in this study system.   

Reviewer 4 Report

Comments and Suggestions for Authors

The authors in the manuscript titled “Effect of nitrogen addition rate on the relationships between multi-dimensional plant diversity and ecosystem production in a temperate steppe” have studied the effect of different N addition doses on the relationships between plant diversity metrics and ecosystem biomass production in a temperate steppe. The topic of this study is relevant to the journal’s readership. The study is interesting, and it discusses an important aspect as the impact of dosage dependent N deposition is less known. Overall, the structure of the manuscript is good.  Some of the comments are as follow

Abstract needs to be more clearly explaining the main results and their impactful meanings. The methodology lacks the information about the soil structure as well as nutrient analysis. As a lot of discussion revolves around the soil dynamics as reasoning for the acquired results so provision of soil analysis with different parameters will improve the authenticity of discussion. Conclusion needs to be more focus and clear.

Comments on the Quality of English Language

 Quality of English language is fine

Author Response

Comments and Suggestions for Authors

The authors in the manuscript titled “Effect of nitrogen addition rate on the relationships between multi-dimensional plant diversity and ecosystem production in a temperate steppe” have studied the effect of different N addition doses on the relationships between plant diversity metrics and ecosystem biomass production in a temperate steppe. The topic of this study is relevant to the journal’s readership. The study is interesting, and it discusses an important aspect as the impact of dosage dependent N deposition is less known. Overall, the structure of the manuscript is good.  Some of the comments are as follow

Abstract needs to be more clearly explaining the main results and their impactful meanings. The methodology lacks the information about the soil structure as well as nutrient analysis. As a lot of discussion revolves around the soil dynamics as reasoning for the acquired results so provision of soil analysis with different parameters will improve the authenticity of discussion. Conclusion needs to be more focus and clear.

Author’s responses: Thank you for your positive comments and constructive suggestions. We updated the Abstract and the Conclusion to be more focus and clear in the revision (Page 2, Lines 32-39, 54-56; Page 22, Lines 703-726). In addition, we did not provide the soil property analysis in the main text, although in the discussion we used soil dynamics as one possible reason for plant community composition change. This is because the effect of N addition rates on the soil physical and chemical properties and its relative to plant community composition change in the same experiment have been published (Tian et al., 2015 and Tian et al., 2022; cited in the Discussion Page 16, Lines 522-526). So, soil dynamics which has been reported by previous studies in the same experiment was used to explain our acquired results under no provision of soil analysis in this study, although we acknowledge that the explanation power by referring other’s study is less than the direct evidence.